# Enhanced Power Generation by Piezoelectric P(VDF-TrFE)/rGO Nanocomposite Thin Film

**DOI:** 10.3390/nano13050860

**Published:** 2023-02-25

**Authors:** Hafiz Muhammad Abid Yaseen, Sangkwon Park

**Affiliations:** Department of Chemical and Biochemical Engineering, Dongguk University, 30 Pildong-Ro 1 Gil, Jung-Gu, Seoul 04620, Republic of Korea

**Keywords:** piezoelectric nanogenerator (PENG), nanocomposite thin film, Langmuir-Schaefer (LS) technique, P(VDF-TrFE), rGO, enhanced performance

## Abstract

In this study we fabricated a piezoelectric nanogenerator (PENG) of nanocomposite thin film comprising a conductive nanofiller of reduced graphene oxide (rGO) dispersed in a poly(vinylidene fluoride-co-trifluoroethylene) (P(VDF-TrFE)) matrix that was anticipated to show enhanced energy harvest performance. For the film preparation we employed the Langmuir-Schaefer (LS) technique to provide direct nucleation of the polar β-phase without any traditional polling or annealing process. We prepared five PENGs consisting of the nanocomposite LS films with different rGO contents in the P(VDF-TrFE) matrix and optimized their energy harvest performance. We found that the rGO-0.002 wt% film yielded the highest peak-peak open-circuit voltage (V_OC_) of 88 V upon bending and releasing at 2.5 Hz frequency, which was more than two times higher than the pristine P(VDF-TrFE) film. This optimized performance was explained by increased β-phase content, crystallinity, and piezoelectric modulus, and improved dielectric properties, based on scanning electron microscopy (SEM), Fourier transform infrared (FT-IR), x-ray diffraction (XRD), piezoelectric modulus, and dielectric property measurement results. This PENG with enhanced energy harvest performance has great potential in practical applications for low energy power supply in microelectronics such as wearable devices.

## 1. Introduction

Poly(vinylidene fluoride) (PVDF) and its copolymers such as poly(vinylidene fluoride-co-trifluoroethylene) (P(VDF-TrFE)), have potential for self-powered flexible electronic devices due to their properties such as thermal stability, mechanical strength, resistance to acids and halogens, light weight, and flexibility, as well as their ferroelectric, dielectric, and piezoelectric properties. Mechanical energy from human body motion can be harvested using piezoelectric, pyroelectric, or triboelectric materials through piezoelectric nanogenerators (PENGs), pyroelectric nanogenerators (PyENGs), and triboelectric nanogenerators (TENGs), respectively [1,2,3]. Among them, the PENGs, which transform mechanical energy into electricity, are advantageous because they are known to have superior energy performance due to high effectiveness and efficiency in harvesting energy from environmental situations such as human motion and structural vibration [4], and thus they can be integrated with device components such as capacitors and rechargeable batteries to store energy with high efficiency.

To improve their piezoelectric performance in microelectromechanical systems (MEMS), many studies have employed a nanocomposite approach that usually mixes nanofillers such as piezoelectric ceramics, semiconductors, and conductive carbon-based materials in the piezoelectric polymer matrix [5,6]. Most commonly used conductive carbon-based materials are carbon nanotube, carbon black, graphene (Gr), graphene oxide (GO), and reduced graphene oxide (rGO) [7,8,9,10,11]. These are known to have many desirable properties. For example, a single-layer of Gr has high intrinsic electron mobility, surface area, thermal conductivity, and electrical conductivity [12,13]. In addition, the higher surface area of Gr is known to provide the piezo-polymer matrix with more polar phases [12,13]. Although rGO is basically a carbonaceous material with similar properties to graphene (Gr), it is advantageous because oxygen-containing functional groups in rGO improve crystallinity by aligning the fluorine atoms at one side in the PVDF and/or P(VDF-TrFE) matrix [14,15]. In addition, rGO shows higher crystallinity than GO due to the formation of sp^2^ hybridization and homogeneous dispersion in the polymer matrix [16,17].

Another important factor for improved piezoelectric performance of PVDF and its copolymers is the content of the β-phase because the β-phase is known to have the best piezoelectric property with a highly polar phase among all crystalline phases of the piezoelectric polymers [7,18,19,20,21]. Up to now, many studies have improved piezoelectric performance by increasing the content of the β-phase with various approaches. Levi et al. reported that the content of the β-phase reached a maximum value with a threshold amount of conductive nanofillers and then declined due to the aggregation of the nanofillers within the matrix [22]. Alamusi et al. [23] found that a drop-casted composite film of rGO in the PVDF matrix showed a maximum output voltage (4 V) at a content of 0.05 wt% rGO after a drawing (60 °C, 10 mm/min) and polling treatment (60 MV/m, 60 min). A self-poled PENG of solution-casted composite film comprising Fe-doped rGO/PVDF was reported to yield an output voltage of 5.1 V and a current of 0.25 μA [24]. A composite film of AlO-rGO/PVDF was reported to generate a high output voltage of 36 V and a current of 0.8 μA, which was ascribed to the role of AlO-rGO as a nucleating agent for β-phase formation [25]. Chen et al. reported that the β-phase in PVDF homopolymer thin films was directly formed by the Langmuir-Blodgett (LB) film deposition technique, and they explained that the molecular chain and dipoles were aligned parallel and perpendicular, respectively, to the substrates, and the dipole orientation and the β-phase formation were attributed to the hydrogen bonds among the PVDF and water molecules through LB deposition [26]. The Langmuir-Schaefer (LS) technique is a deposition method modified from the LB technique, and transfers a monolayer film from the air/water interface to the solid substrate by horizontal deposition, whereas the LB technique uses a vertical deposition. These techniques are known to yield uniform, highly ordered, and molecularly well-defined thin films on solid substrates [27]. In addition, thin films prepared from these techniques are known to have high β-phase content and crystallinity, which enable high piezoelectric performance [27,28]. Therefore, these techniques have advantages compared to other film formation techniques of PVDF and its copolymers’ nanocomposite, such as thermal imprinting [29], spin coating [9], nano-confinement [30], and drop-casting methods [31]. Overall piezoelectric performance, like the output open-circuit voltage (V_OC_) and short-circuit current (I_SC_) of PENGs using carbon-based nanofillers and PVDF, and/or its copolymers, is known to be low and restrain their practical use in electrical appliances [32]. Therefore, there should be effective ways to fabricate PENG devices with improved performance.

In this work, we employed a nanocomposite thin film comprising conductive rGO NPs in a P(VDF-TrFE) matrix to fabricate PENG with enhance energy harvesting performance. A thin film of multiple layers was prepared by repeated deposition of nanocomposite monolayers spread at the air/water interface onto indium tin oxide-coated polyethylene terephthalate (ITO-PET) substrates by the LS technique. A PENG was prepared by arranging the thin film-coated ITO-PET substrates into a sandwich structure. We optimized the PENG’s piezoelectric performance by investigating the effect of rGO content, surface pressure, and number of layers. The optimized piezoelectric performance is discussed in terms of β-phase content, crystallinity, and the dielectric constant of thin films.

## 2. Experimental Section

### 2.1. Materials

Poly(vinylidenefluoride-co-trifluoroethylene) (P(VDF:TrFE)= 70:30 mol%) and reduced graphene oxide (rGO, research grade powder, surface area of 103 m^2^/g) were used as the polymer matrix and the nanofiller, respectively. An indium tin oxide-coated polyethylene terephthalate (ITO-PET) sheet (8 cm × 2.5 cm) was used as a solid substrate for film deposition by the LS technique. *N*,*N*-dimethylformamide (DMF, 99.8%) and acetone (99.7%) were used as solvents. All chemicals were purchased from Sigma-Aldrich (St. Louis, MO, USA) and used as received without further purification.

### 2.2. Preparation of P(VDF-TrFE)/rGO Solution and LS Thin Film

A nanocomposite solution of 50 g was prepared by dissolving 25 mg of P(VDF-TrFE) and dispersing a specific amount of rGO in a mixed solvent of DMF/acetone (*v*/*v* 40/60) 59 mL. Four solutions were prepared by dispersing different amount of rGO in the P(VDF-TrFE) solution, which were P(VDF-TrFE) solutions containing 0.002, 0.004, 0.02, and 0.04 wt% rGO, and then they were sonicated for 15 min. A clean Langmuir trough of Teflon was filled with deionized water (18 MΩ·cm), and the P(VDF-TrFE)/rGO solution was spread on the air/water interface at room temperature (23 °C) using a microsyringe of 500 μL. A monolayer of P(VDF-TrFE)/rGO was compressed to a target surface pressure by moving two barriers inward at 10 mm/min, and was horizontally transferred to the ITO/PET surface. A thin film of multilayers was prepared by repeating the monolayer deposition procedure five times. The thin film was then dried in the air at room temperature and stored in a desiccator for further analysis and/or experiments. This procedure of thin film preparation is illustrated in Figure 1a.

### 2.3. Fabrication of PENG Device

A PENG device was fabricated from the LS film. The film deposition area (about 6 cm × 2.5 cm) of two ITO-PET sheets were cut and placed together touching each other at the coating area. To make a sandwich structure, two new ITO-PETs were placed on the top of the coating area at each side. The ITO coatings of the two ITO-PETs (about 2 cm × 2.5 cm) were positioned to protrude to the opposite side and were used as electrodes connected with alligator clips. Figure 1b shows a picture of the PENG device and an illustration of the sandwich structure. Nonconductive adhesive tape was wrapped tightly around the device to prevent any breakage during bending and releasing.

### 2.4. Characterization

Surface morphology and microstructure were observed using a scanning electron microscopy (SEM, JSM-6700F, JEOL Ltd., Tokyo, Japan). To examine the crystalline structure of the thin film, an X-ray diffractometer (XRD, D/Max2500, Rigaku, Tokyo, Japan) and FT-IR spectroscopy (Nicolet iS50, Thermo Fisher Scientific, Waltham, MA, USA) were used. The XRD was operated with Cu target radiations in the 2θ range of 10 to 30° with a step width of 0.02° and a scan speed of 1.0°/min. A quasi-static d_33_ m (Han Tech, PM 3500, Seoul, South Korea) was used to measure the average d_33_ constant after taking five measurements for each sample. A LCR meter (HIOKI 3532 LCR HITESTER, Plano, TX, USA) was used to calculate dielectric properties using a two-point probe system. The output voltage generated by the PENG was measured using a functional digital storage oscilloscope (EDUX1002G) at 1 and 2.5 Hz bending and releasing frequencies using a motorized bending machine Exi-Servo (EzS-NDR-42L-A-D).

## 3. Results and Discussion

### 3.1. Characteristics of Nanocomposite Monolayer

As the monolayer is compressed, the area (A) of spread monolayer decreases with almost same number of slowly soluble P(VDF-TrFE) molecules and rGO particles on the surface. Thus, the distance among the molecules and the particles decreases and the surface pressure (π) increases due to a higher repulsive intermolecular force among monolayer molecules and/or particles. This relationship between π and A is usually measured at a constant temperature during the compression, which is called the surface pressure-area (π-A) isotherm. Essential information concerning the nanocomposite monolayer can be acquired from the surface pressure-area (π-A) isotherm, and morphological properties of the film are usually affected by the deposition conditions [33]. Figure 2 shows the π-A isotherms of spread monolayers of five samples. As shown in Table 1, as the rGO content increased from 0 to 0.02 wt%, the surface area increased from 79.6 to 106.9 cm^2^ (34.3%) at π = 5 mN/m and from 69.7 to 102.4 cm^2^ (46.9%) at π = 15 mN/m, respectively. This suggests that the more rGO particles disperse in the unit area of monolayer, the higher the repulsive intermolecular force exerted among P(VDF-TrFE) molecules and rGO particles. More surface area was taken by a higher content of rGO because of the higher repulsive force among molecules and particles. It should be noted that a spread monolayer of P-rGO-0.001 was also examined and yielded a π-A isotherm undistinguishable from that of pristine P(VDF-TrFE), within error. We presumed that a smaller rGO content than 0.002 wt% did not affect monolayer properties significantly, and thus other film properties, so we did not investigate other characteristics and piezoelectric performance of this thin film further.

### 3.2. Morphology Analysis by SEM

Morphology and crystalline structure were observed with a SEM. Figure 3 shows SEM images for the thin-film samples of P-rGO-0, P-rGO-0.002, P-rGO-0.004, P-rGO-0.02, and P-rGO-0.04. As seen in Figure 3, P-rGO-0.002, P-rGO-0.004, and P-rGO-0.02 showed some aggregations of less than one micron, but P-rGO-0.04 showed much larger particles with the size of a few microns (see the circles in the figure), which were presumably aggregations of rGO particles, whereas P-rGO-0 formed a film of better quality with high uniformity and homogeneity. This trend was supported by the π-A isotherm results. The π-A isotherms in the Figure 2 can be classified into three groups according to their relative position. The first group (P-rGO-0) was positioned in the left, the second one (P-rGO-0.04) was positioned in the right, and the third group (P-rGO-0.002, P-rGO-0.004, and P-rGO-0.02) was positioned in the middle. The SEM morphology results suggest that the thin film start to form large aggregates of micron dimensions at a rGO content of 0.04 wt%, presumably due to agglomeration of rGO particles. In this study, the P-rGO-0.04 sample was expected to provide lower piezoelectric performance because nonuniformity and heterogeneity in the film have been reported to reduce piezoelectric efficiency in the literature [34].

### 3.3. FT-IR Analysis

FT-IR was used to analyze the crystalline phase of the P(VDF-TrFE) matrix and the interaction between rGO particles and P(VDF-TrFE) molecules. Figure 4 shows the FT-IR spectra of five thin films with detailed peaks in the range of 750–1500 cm^−1^, and the analysis of the β phase content (%).

As seen in the Figure 4a, there were several distinct peaks in the FT-IR spectra. There were stretching vibration peaks of –CH_2_– around 2900 cm^−1^, and peaks for P(VDF-TrFE) crystalline phases in the range of 750 to 1500 cm^−1^. Table 2 shows characteristics of the latter peaks. Among those peaks, the peak by symmetric stretching of CF_2_ and CC was reported to have the β-phase, whereas the peak by asymmetric stretching of CC, and wagging of CH_2_ and CF_2_, was the α-phase [35,36,37,38]. These characteristics can be used to calculate the percent content of the β-phase. The percent β-phase content, F(β) can be calculated by the following equation [39]:(1)$$F\left(\beta \right)=\frac{{\;A}_{\beta }}{\left(\frac{K_{\beta }}{K_{\alpha }}\right){\;A}_{\alpha }+{\;A}_{\beta }}\;\times 100$$
where $\;A_{\beta }$ and $\;A_{\alpha }$ are the absorption intensities at 846 and 761 cm^−1^, respectively, and $\;K_{\beta }$ and $\;K_{\alpha }$ are the absorption coefficients of 7.7 × 10^4^ and 6.1 × 10^4^ cm^2^ mol^−1^, for the β-phase and the α-phase, respectively. The calculation results are shown in the Figure 4b (see the insert).

With a 0.002 wt% rGO concentration, the β-phase percent content showed a maximum of 98%, and then decreased to 87% as the rGO concentration increased up to 0.04 wt%. This implies that at the concentration of rGO 0.002 wt% the P(VDF-TrFE) polymer chains are mostly oriented parallel to the substrate [40]. A tiny absorption peak around 3400 cm^−1^ suggests intermolecular hydrogen bonding between the P(VDF-TrFE) chain, –CH_2_– dipoles, and oxygen-containing functional groups of rGO. The stretching vibration peaks of –CH_2_– around 2900 cm^−1^ shifted towards lower frequency region, which indicated stronger dipolar interactions of –CH_2_–CF_2_– in the polymer matrix. These two attractive forces of hydrogen bonding and dipolar interaction helped dipole orientation in P(VDF-TrFE) chains in the nanocomposite film of P(VDF-TrFE)/rGO [41,42]. This result of optimum rGO content can be explained in terms of interaction between surface charge due to oxygen-containing functional groups such as C–O, C=O, COO–, OH and delocalized π electrons of rGO and –CH_2_– CF_2_ dipoles in polymer chains [15,43]. This interaction accommodates the P(VDF-TrFE)/rGO nanocomposite film to form the β-phase easily. With a specific content of rGO, the interaction between surface charge of rGO and dipoles in polymer chains maximizes all-trans (TTTT) molecular conformation (i.e., the β-phase in the polymer matrix), but with a higher content of rGO it results in rGO particles oppositely aligned to the chain alignment, thus reducing β-phase development [24,44]. The β-phase of all-trans molecular conformation in the polymer matrix might be further enhanced by the preparation method in this study. As an explanation, the HO-H groups in the water subphase help build hydrogen bonds with C-F groups of the P(VDF-TrFE) chain in the spread monolayer, and thus the P(VDF-TrFE) molecules should be oriented with C-F groups inward to the water subphase [26], which is similar to the all-trans molecular conformation in the β-phase. This explanation is illustrated in Figure 4c. 

### 3.4. XRD Analysis

An XRD was used to analyze the crystal structure of the nanocomposite thin films. Figure 5a shows XRD patterns for five nanocomposite thin films. All the patterns showed distinct peaks at about 2θ = 19.7, corresponding to the (110/220) planes, recognized as polar β-phases [45]. It is noted that the peak with the rGO concentration of 0.002 wt% was distinctly larger than those with other concentrations. This trend can be explained in terms of degree of crystallization due the interaction between rGO and polymer chains. At the rGO concentration of 0.002 wt%, there seemed to be a proper degree of interaction between rGO and the polymer chains, leading to optimum β-phase with the highest order and crystallinity in the polymer matrix. However, the presence of rGO more than the optimum amount triggered aggregation by creating more nucleation seeds, which reduced the formation of the β-phase crystal structure. Importantly, the increase in the β-phase diffraction peak intensity was significantly influenced by uniform and homogeneous distribution of rGO in the polymer matrix.

The slight peak shift that took place at 0.002 wt% was ascribed to the change in lattice parameter of P(VDF-TrFE). The crystallinity of thin film appeared to be increased compared to the pristine P(VDF-TrFE) thin film. The percentage crystallinity (*X_C_*) was calculated using the following equation [35]:(2)$$X_{C}\;\left(\%\right)=\frac{\sum {\;A}_{Crys}}{\sum A_{Crys}+\sum A_{Amor}}\;\times 100$$
where ∑*A_Crys_* and ∑*A_Amor_* are the total integral area under the crystalline and amorphous zones, respectively. As expected, the P-rG-0.002 showed a maximum degree of crystallinity of 56%, and P-rGO-0.04 a minimum of 25% (see Table 3). Similar trends of XRD pattern and crystallinity have been reported with a spin-coated P(VDF/TrFE)/rGO film in the literature [44].

### 3.5. Piezoelectric Modulus and Dielectric Properties

The piezoelectric modulus (d_33_) for five thin films of ten nanocomposite monolayers was measured by employing a quasi-static d_33_ m at a constant force of 250 × 10^−3^ N. The results are shown in Figure 5b. Among the five films, P-rGO-0.002 yielded the highest d_33_ value of 98 (±2.2) *pC*/*N*, which is consistent with previous results such as β-phase content and crystallinity. According to the principle of the piezoelectric effect, an applied external force produces an electric potential field in the longitudinal direction, which is nonuniformly distributed in the nanocomposite thin film due to different dielectric properties of rGO and P(VDF-TrFE) matrix (see Appendix A). These results indicate that P-rGO-0.002 had the highest degree of additional potential field in the film because it is known that additional lateral potential field increases the piezoelectric modulus (d_33_) [46].

The dielectric constant (ε) and dielectric loss (tan δ) were measured for five thin films at room temperature using a two point-probe LCR meter in the AC signals frequency range of 10^4^–10^6^ Hz. The ε and tan δ values were calculated from the measured capacitance (*C*) using the following equation [47]:(3)$$\varepsilon =\frac{C\;d}{\varepsilon ˳\;A}$$
where ε˳, *d*, and *A* denote dielectric constant of a vacuum (8.85 × 10^−12^ Fm^−1^), the distance (i.e., film thickness), and the surface area of the electrode, respectively [47]. At 10^4^ Hz frequency, the dielectric constant value was 17 for the pristine P(VDF-TrFE) film and reached a maximum value of 21 at a rGO concentration of 0.002 wt%. As shown in Appendix A, this trend can be explained by a model of micro-capacitor formation [43]. In the model, the nanocomposite thin film with the optimum amount (0.002 wt%) of rGO shows higher dielectric constant than that for any other films because rGO particles form proper number of micro-capacitors. In other words, two neighboring rGO platelets act as electrodes, and the polymer P(VDF-TrFE) as a dielectric medium in the thin films. At the optimum concentration of rGO, a proper number of micro-capacitors are formed to increase the film capacitance with an enhanced dielectric constant by aligning the dipoles. However, above the optimum amount of the rGO concentration, the rGO particles start to aggregate, forming an electrically conductive percolated system, thus reducing dielectric properties. Moreover, a further increase in filler content creates loss of insulating property and the charge carriers are accumulated at the interfaces between filler and polymer [48,49]. According to studies in the literature, the higher ε value may be due to dipoles having enough time to align at lower frequencies (10^4^). At higher frequencies (10^6^ Hz), the dipoles do not have enough time for alignment with the applied AC field direction. Thus, the dielectric constant becomes reduced. The dielectric loss (tan δ) followed the same trend as dielectric constant (ε), as shown in Appendix A. The highest dielectric loss at the rGO concentration of 0.002 wt% was ascribed to formation of more conductive pathways. The low value of tan δ might be due to polarization loss, conduction loss, and produced heat after colloidal formation of dipoles at high AC field frequencies [46]. Table 4 summarizes all the characteristics of the five nanocomposite thin films.

### 3.6. Energy Harvesting

Five different PENG devices with a sandwich structure and thin films of ten nanocomposite multilayers of different rGO content were subjected to the open-circuit voltage (V_OC_) and short-circuit current (I_SC_) measurements. An oscilloscope was used for the measurements when the devices were tested with different bending and releasing frequencies of 1 Hz and 2.5 Hz, and a bending angle of 50°. As a result of bending and releasing, a piezoelectric potential is generated at the bottom and top electrodes of the PENG devices. Compressive deformation vanishes upon releasing the bending force; hence, the generated electrons flow back to regain the potential, which causes the voltage to change in the opposite direction. Therefore, positive and negative output voltages are generated upon bending and releasing a PENGs, and the current has the same increasing and decreasing trend as the output voltage. Figure 6 shows V_OC_ signals and average peak-peak V_OC_ of five PENGs at π = 5 and 15 mN/m. As seen in Figure 6a,b, the PENG with ten layers of P-rGO-0.002 showed the highest peak-peak V_OC_ of 88 V at 2.5 Hz and π = 5 mN/m, whereas that with P-rGO-0 was about half the V_OC_ at 42 V under the same conditions. The PENGs with same number of layers of P-rGO-0.004, P-rGO-0.02, and P-rGO-0.04 showed peak-peak V_OC_ values of 74, 36 and 34 V, respectively. The decrease in the peak-peak V_OC_ value with increased rGO concentration for P-rGO-0.004, P-rGO-0.02 and P-rGO-0.04 PENGs can be ascribed mainly to the aggregation of rGO. More aggregation of nanofiller particles has been known to decrease the flow rate of electrons by reducing the effective electrostatic fields in between nano-flakes and polymer molecules [45]. In addition, the large surface area of rGO may accelerate the generation of CO dipoles, which increases micro-capacitor formation to hamper piezoelectric performance, as discussed previously. To check the contribution of ITO-PET substrate to voltage creation, V_OC_ for the ITO-PET was measured under the same conditions, but no significant signals were detected.

When the bending and releasing frequency was reduced to 1 Hz, the peak-peak V_OC_ values decreased to lower values of less than half. When the surface pressure increased to π = 15 mN/m, the peak-peak V_OC_ values for P-rGO-0.004, P-rGO-0.02, and P-rGO-0.04 were not significantly changed within error (less than 10%), whereas that for P-rGO-0 significantly increased (more than 50%), and that for P-rGO-0.002 decreased (about 20%). This indicates that the surface pressure was not a critical factor influencing piezoelectric performance. Integration of all these piezoelectric performance test results indicates that the thin film of ten nanocomposite monolayers with 0.002 wt% rGO deposited π = 5 mN/m yielded the best piezoelectric performance, and thus this film is called ‘optimum’ or ‘optimized’ thin film hereafter.

As shown in Figure 7a, the short-circuit I_SC_ values showed similar trends to V_OC_. In addition, Appendix A show similar trends of peak-peak V_OC_ values for other film thicknesses with different number of layers. As shown in Figure 7b, the thicker film (as much as 40 nm (i.e., thickness of ten layers)) [50] yielded the larger V_OC_ values. This result suggests that a more conductive network is formed with a thicker film, and the aggregates in the network presumably have a less negative effect on the piezoelectric network in the thicker film.

For a practical application, resistance dependencies of voltage and current were investigated by measuring voltage (V_L_) as a function of resistance load and by calculating current (I_L_) in a short-circuit. Figure 7c shows the V_L_ and I_L_ results for the optimum thin film PENG. With increasing resistance, the V_L_ steadily increased, and the I_L_ decreased, and the rate of increase and decrease became smaller around the inflection point of R_L_ = 20 MΩ. Figure 7d shows the power density profile as a function of resistance with a maximum of 16.5 μW/cm^2^ at R_L_ = 20 MΩ, which was calculated using the following equation:(4)$$P=\frac{V{˻}^{2}}{R˻}$$

The optimum PENG with high power density (*P*) can store the electrical potential in conjunction with different capacitors for practical use. To demonstrate feasibility of practical use of the optimum PENG, a simple circuit was designed to comprise several light emitting diodes (LEDs), a capacitor, and four rectifiers, as shown in the Figure 8a. Using the circuits with different capacitors (0.1, 0.22, 1, and 10 µF), the electrical energy could be successfully stored, and the LEDs could be lit as shown in the Appendix A. As shown in the potential-time and energy storage-time profiles (Figure 8b,c), the maximum voltage of 65 V (DC) could be stored using a 0.1 µF capacitor, and with a 1 µF capacitor the maximum energy storage was calculated to be 999 µJ by the formula E_C_ =$\;\frac{1\;}{2}$ CV^2^ where V and C denote the voltage and the capacitance, respectively [51]. The energy generated from the optimized PENG can be used to drive a low power- consuming electronics such as wearable device. As such, compared to conventional PENGs reported in the literature, this optimum PENG with enhanced performance showed several distinct advantages, such as using a minimal amount of rGO nanofiller, the capability of coating a large area (LS film), a facile and cheap fabrication process, and superior energy harvesting performance. Table 5 compares characteristics and performance of the optimum PENG with those of PVDF-based PENGs reported in the literature. These advantages of the optimum PENG make it a promising candidate for practical applications of low energy power supply in microelectronics, such as wearable devices.

## 4. Conclusions

A piezoelectric nanogenerator (PENG) of nanocomposite thin film comprising rGO particles dispersed in a P(VDF-TrFE)) matrix was fabricated by arranging the film coated ITO-PET substrates in a sandwich structure. The nanocomposite thin film was prepared by the Langmuir-Schaefer (LS) technique to maximize the content of the polar β-phase without any traditional polling or annealing process. Several PENGs with different rGO contents, surface pressures, and different film thickness were subject to SEM observation, FT-IR spectroscopy, XRD, piezoelectric modulus, dielectric properties, and piezoelectric properties, such as open-circuit output voltage (V_OC_) measurements, to optimize film composition, density, and thickness. A PENG with a film of ten layers of nanocomposite monolayer containing 0.002 wt% rGO at π = 5 mN/m was determined to have optimal performance of peak-peak V_OC_ of 88 V, energy storage of 999 µJ with 1 µF capacitor. These results are discussed in terms of the β-phase, crystallinity, piezoelectric modulus d_33_, and dielectric properties. The energy generated from the optimum PENG can be used to directly drive low-power-consuming electronics such as wearable devices. Importantly, the novel optimum PENG fabricated in our study revealed several advantages, including superior energy harvesting performance with a minimal amount of rGO nanofiller, the capability of coating a large area, and a facile and cheap fabrication process. Therefore, these advantages of the optimum PENG make it a promising candidate for practical applications of low energy power supply in microelectronics.

## Figures and Tables

**Figure 1 nanomaterials-13-00860-f001:**
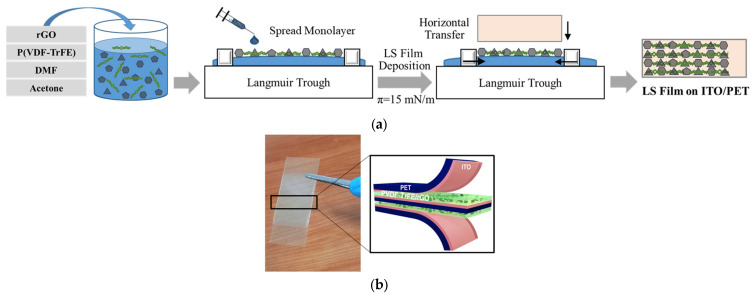
(**a**) Thin-film preparation procedure; (**b**) picture of fabricated PENG with a sandwich structure.

**Figure 2 nanomaterials-13-00860-f002:**
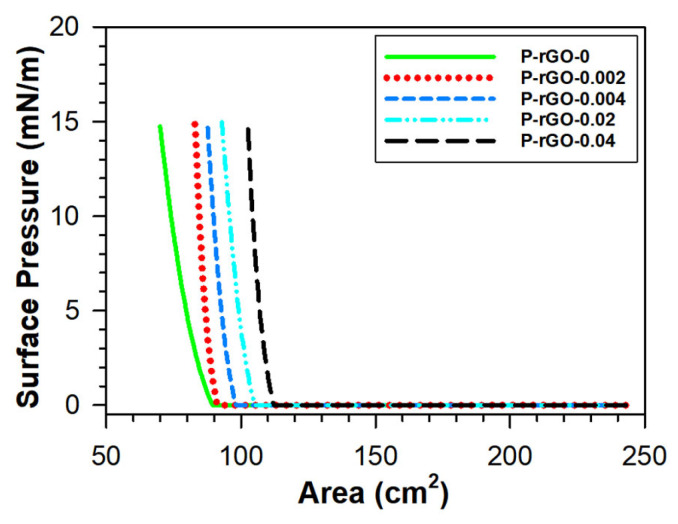
Surface pressure-area (π-A) isotherms of nanocomposite monolayers.

**Figure 3 nanomaterials-13-00860-f003:**
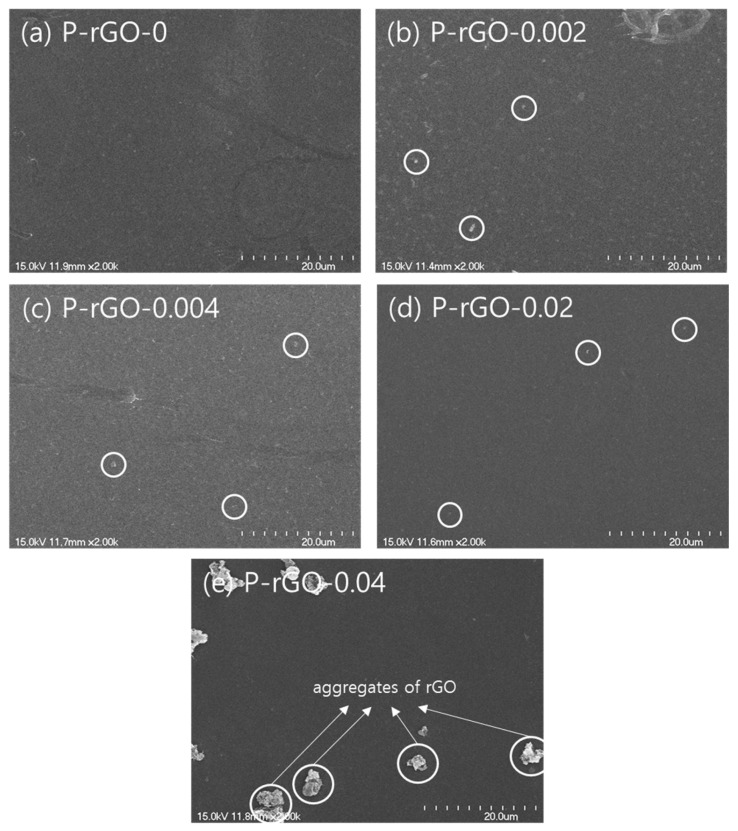
SEM images of (**a**) P-rGO-0, (**b**) P-rGO-0.002, (**c**) P-rGO-0.004, (**d**) P-rGO-0.02, and (**e**) P-rGO-0.04 at π = 5 mN/m.

**Figure 4 nanomaterials-13-00860-f004:**
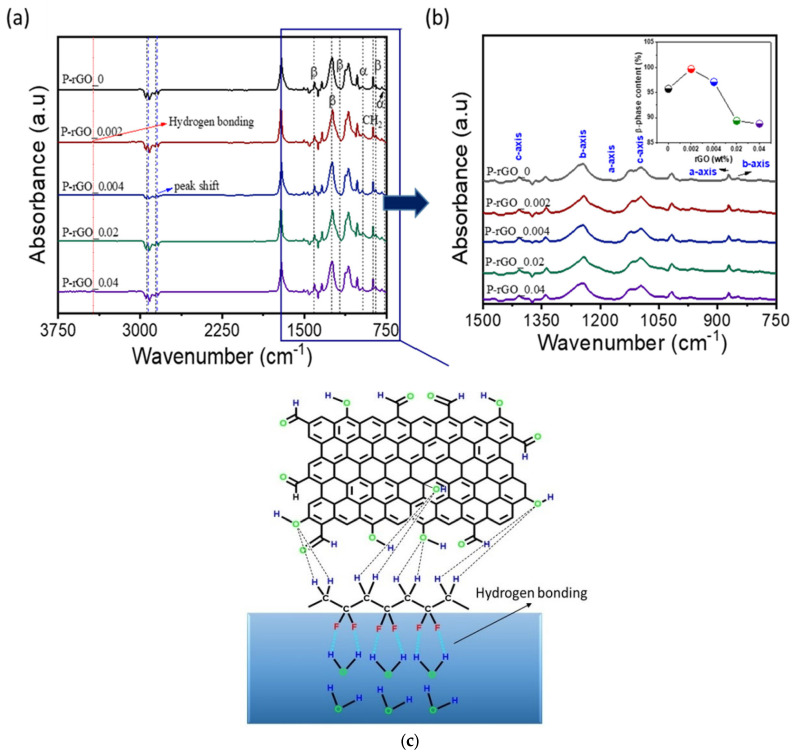
(**a**) FT-IR spectra of five thin films; (**b**) detailed peaks in the range of 750–1500 cm^−1^ in the FT-IR spectra, and β-phase content (%) analysis; (**c**) illustration of hydrogen bonding among water molecules, polymer matrix, and rGO particles.

**Figure 5 nanomaterials-13-00860-f005:**
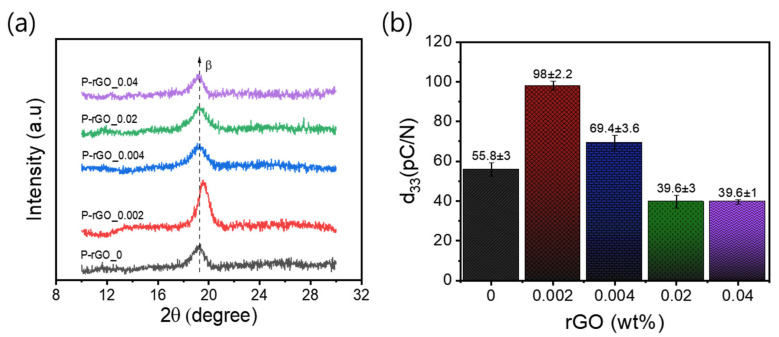
(**a**) XRD patterns (2θ = 10–30°) for five thin films; (**b**) piezoelectric modulus (d_33_) plot as a function of rGO content.

**Figure 6 nanomaterials-13-00860-f006:**
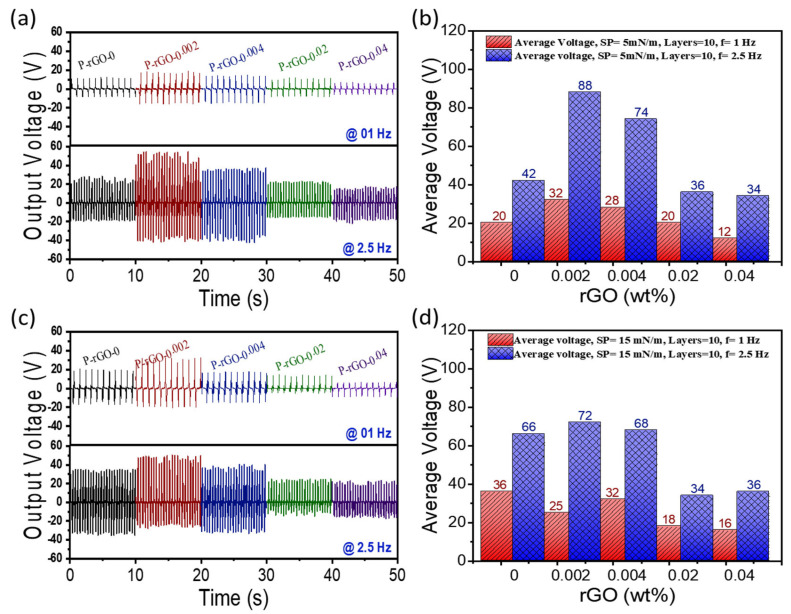
(**a**) V_OC_ signals generated by five PENGs at π = 5 mN/m; (**b**) average peak-peak V_OC_ of five PENGs at π = 5 mN/m; (**c**) V_OC_ signals generated by five PENGs at π = 15 mN/m; (**d**) average peak-peak V_OC_ of five PENGs at π = 15 mN/m.

**Figure 7 nanomaterials-13-00860-f007:**
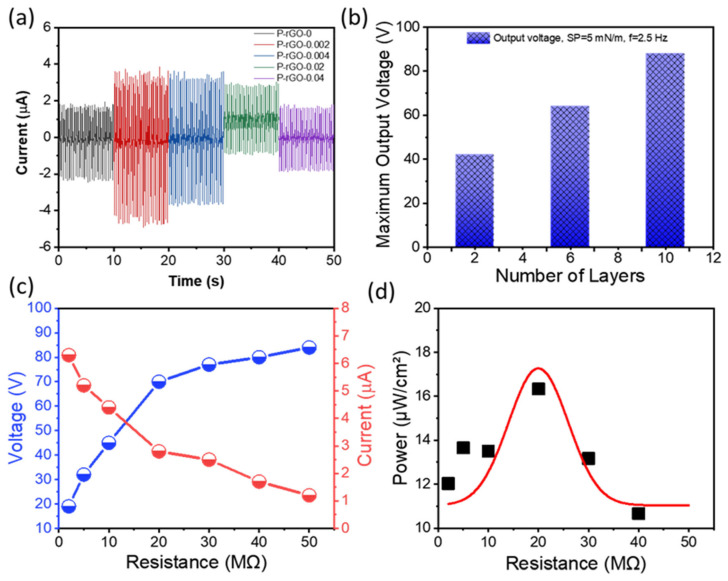
(**a**) I_SC_ of the PENGs with thin films of ten monolayers with different rGO contents at π = 5 mN/m and 2.5 Hz; (**b**) peak-peak V_OC_ for the PENGs with thin films of different number of monolayers with 0.002 wt% rGO at π = 5 mN/m and 2.5 Hz; (**c**) resistance dependencies of voltage and current; (**d**) resistance dependency of power density for the optimum PENG.

**Figure 8 nanomaterials-13-00860-f008:**
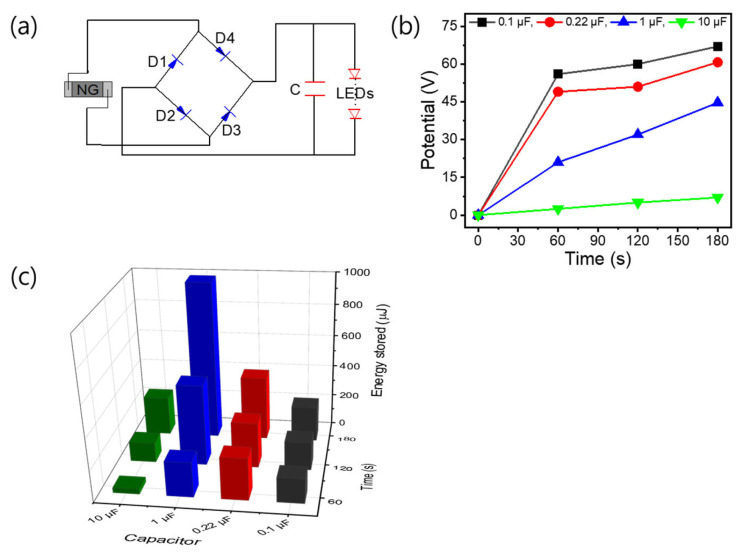
(**a**) Simple circuit design with a capacitor, four rectifiers, and several LEDs to demonstrate the feasibility of the optimum PENG; (**b**) potential-time profiles obtained by the circuits with different capacitors at 2.5 Hz; (**c**) energy storage-time profiles by the circuits with different capacitors.

**Table 1 nanomaterials-13-00860-t001:** Surface area at π = 5 and 15 mN/m as a function of rGO content.

Sample	P-rGO-0	P-rGO-0.002	P-rGO-0.004	P-rGO-0.02	P-rGO-0.04
A (cm^2^) at π = 5 mN/m	79.6	86.8	92.6	99.1	106.9
A (cm^2^) at π = 15 mN/m	69.7	82.8	87.5	92.9	102.4

**Table 2 nanomaterials-13-00860-t002:** Characteristics of P(VDF-TrFE) phase absorption peaks in the FT-IR spectra.

Absorption Peak (cm^−1^)	Vibrational Modes	Phase	References
761	_b_CH_2_	α	[35]
846	𝜈_σ_ CF_2_ + 𝜈_σ_CC	β	[36,37]
880	rCH_2_ − 𝜈_a_CF_2_ − rCF_2_	β	[36,37]
1095	𝜈_a_CC + _w_CH_2_ − _w_CF_2_	β	[36,38]
1177	𝜈_a_CF_2_ − rCF_2_	β	[36,37]
1408	_w_CH_2_ − 𝜈_a_CC	β	[36,37]

(𝜈_a_) asymmetric stretching, (𝜈_σ_) symmetric stretching, (b) bending, (_w_) wagging, (r) rocking. The sign “−” and “+” indicate the phase relation between the symmetry coordinates.

**Table 3 nanomaterials-13-00860-t003:** Degree of crystallinity (%) calculated from XRD patterns of five thin films.

Sample	Percentage Crystallinity (%)
P-rGO-0	35
P-rGO-0.002	56
P-rGO-0.004	36
P-rGO-0.02	29
P-rGO-0.04	25

**Table 4 nanomaterials-13-00860-t004:** Summarized characteristics of five thin films.

Sample	β-Phase (%)	Crystallinity (%)	d_33_ (pC/N)	Dielectric Constant (ε)	Dielectric Loss (tan δ)
P-rGO-0	95	35	55.8	17	2.4
P-rGO-0.002	98	56	98	21	1.1
P-rGO-0.004	96	36	69.4	20	1.6
P-rGO-0.02	89	29	39.6	07	2.9
P-rGO-0.04	87	25	39.6	05	3.2

**Table 5 nanomaterials-13-00860-t005:** Comparison of the optimum PENG in this study with PVDF-based PENGs reported in the literature.

PENG		Coating Method	Form	Size (cm^2^)	Peak-Peak V_OC_ (V)	Year	Ref.
Material	Filler Content (wt%)						
P(VDF-TrFE)/Gr	0.15	solution casting	film	0.1 × 0.3	12.43	2019	[52]
P(VDF-TrFE)/rGO	0.5	spin coating	film	2.5 × 2.5	89.7	2019	[44]
PVDF/rGO	0.8	electrospinning	nanofiber	4 × 5	4.38	2020	[46]
PVDF/GO	0.4	electrospinning	nanofiber	4 × 5	1.15	2020	[46]
P(VDF-TrFE)/rGO	0.1	drop casting	sheet	3 × 2	2.40	2018	[43]
P(VDF-TrFE)/rGO	0.1	scrap coating	film	4 × 5	8.32	2019	[42]
P(VDF-TrFE)/GO	-	drop casting	film	-	4.30	2015	[31]
P(VDF-TrFE)/rGO	0.002	LS depsition	LS film	8 × 2.5	88	2023	This work

## Data Availability

The data in this study are available upon request from the corresponding author.

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
