# Peer review of "Enhanced Power Generation by Piezoelectric P(VDF-TrFE)/rGO Nanocomposite Thin Film"

_nanomaterials, 2023, doi:10.3390/nano13050860_

Round 1
Reviewer 1 Report
A PENG based on P(VDF-TrFE)/rGO composite LS-film has been developed. The paper should be revised based on the following comments:
1. The equation 1 and equation 2 in the manuscript should be cited if they are not deduced by the authors.
2. Are the results in Figure 5 and Figure 6 obtained from the sample P-rGO_0.002 which has a largest d33?
3. In table 5, PVDF-based PENGs open-circuit voltages (VOC) in reported studies and the presented work are provided. For the presented work, peak-peak voltage has been provided. Are the open-circuit voltages in the reported studies also peak-peak? The comparisons should be conducted for the same type voltages.
4. The authors should stress the innovation of the presented work in the section of conclusion.
Author Response
"Please see the attachment"

Reviewer 2 Report
In this manuscript, the authors successfully developed a PENG based on P(VDF-TrFE)/rGO composite LS-film by adding conductive nanofiller in piezo-electric copolymer. But there are still some other issues needed to be solved before considering it to be accepted for publication. The details are listed as follows:
1. The aspect ratios of all Figures are confusing and need to be improved.
2. For the SEM images in Figure 2, rGO cannot be recognized. It is better to provide the EDS results.
3. “In particular, by increasing the rGO wt% at 0.002 in the nano-composite, β-phase diffraction peak intensity increased significantly.” Why? Would you please elaborate on this point?
4. The English is awkward in several places.
Author Response
"Please see the attachment."

Reviewer 3 Report
Comments to the Authors
In this manuscript authors fabricated a piezo-electric nanogenerator (PENG) of P(VDF- 9 TrFE)/rGO nanocomposite Langmuir-Schaefer film (LS-films) by the addition of reduced graphene oxide (rGO) as a conductive nanofiller with superior piezo-electric energy harvesting ability through a simple Langmuir- Schaefer (LS) technique that is in favor of direct nucleation of higher polar β-phase by hydrogen bonding without any traditional polling or annealing process. This research has value for the researchers in the related areas. However, the paper needs improvement before acceptance for publication. My detailed comments are as follow:
1. In the introduction section authors should introduced following relevant articles related to PVDF and reduced graphene oxide with description of properties
a. doi.org/10.1007/s13233-019-7039-y
b. doi.org/10.1016/j.nanoso.2020.100487
2. The objective of manuscript should be brief.
3. There are few typos and grammatical errors,
4. Why authors used rGO instead of other carbon based materials?
5. The quality of the figure 3 should be improved.
Reviewer 4 Report
This paper describes the fabrication of a piezoelectric nanogenerator (PENG) using a P(VDF-TrFE)/rGO nanocomposite Langmuir-Schaefer film. The PENG generated a high open-circuit voltage of up to 88 V through bending and releasing at 2.5 Hz frequency, with the performance being influenced by the crystallinity of the higher electroactive β-phase and the electrostatic interaction among carboxyl functional groups in rGO, −CH2−/−CF2− dipoles of P(VDF-TrFE), and hydrogen bonding interaction. The prepared PENG has potential applications in energy storage devices, flexible nanoelectronic devices, and futuristic research exploration. I recommend this manuscript be published in the journal of " Nanomaterials" with the following modifications.
1. Could the authors provide the performance of samples with rGO ratios between 0 and 0.002 wt%? If not, please provide an explanation.
2. The authors should organize the discussion of the figures in a logical order to avoid duplicate citations and discussions. For example, the description of Fig 6a is found in the paragraph discussing Fig 3.
3. The authors should provide high-magnification SEM images to better support the morphological details of the film, as the current magnification is too low.
4. The authors should carefully check for typos, such as revising "55.8±30" in Table 4.
Round 2
Reviewer 2 Report
The author has addressed all of the original comments and the revised manuscript is suitable for publication in Nanomaterials.
Reviewer 4 Report
This revision has significantly improved the manuscript. I recommend the manuscript for publication.